**Subject Category:**
Biology (whole organism)

behaviour/environmental science/ecology

acoustic disturbance, anthropogenic noise, behavioural response, environmental risk assessment, marine mammal conservation

**Author for correspondence:**
Isla M. Graham
e-mail: i.graham@abdn.ac.uk

# Harbour porpoise responses to pile-driving diminish over time

Isla M. Graham[1], Nathan D. Merchant[2], Adrian Farcas[2], Tim R. Barton[1], Barbara Cheney[1], Saliza Bono[1] and Paul M. Thompson[1]

[1]Lighthouse Field Station, School of Biological Sciences, University of Aberdeen, George Street, Cromarty, Ross-shire IV11 8YL, UK
[2]Centre for Environment, Fisheries and Aquaculture Science (Cefas), Pakefield Road, Lowestoft NR33 0HT, UK

  IMG, 0000-0001-7018-3269; NDM, 0000-0002-1090-0016; AF, 0000-0002-3320-8428; BC, 0000-0003-4534-5582; PMT, 0000-0001-6195-3284

Estimating impacts of offshore windfarm construction on marine mammals requires data on displacement in relation to different noise levels and sources. Using echolocation detectors and noise recorders, we investigated harbour porpoise behavioural responses to piling noise during the 10-month foundation installation of a North Sea windfarm. Current UK guidance assumes total displacement within 26 km of pile driving. By contrast, we recorded a 50% probability of response within 7.4 km (95% CI = 5.7–9.4) at the first location piled, decreasing to 1.3 km (95% CI = 0.2–2.8) by the final location; representing 28% (95% CI = 21–35) and 18% (95% CI = 13–23) displacement of individuals within 26 km. Distance proved as good a predictor of responses as audiogram-weighted received levels, presenting a more practicable variable for environmental assessments. Critically, acoustic deterrent device (ADD) use and vessel activity increased response levels. Policy and management to minimize impacts of renewables on cetaceans have concentrated on pile-driving noise. Our results highlight the need to consider trade-offs between efforts to reduce far-field behavioural disturbance and near-field injury through ADD use.

## 1. Introduction

Recognition of the potential impact of underwater noise disturbance on marine wildlife has resulted in major policy developments affecting the management of offshore activities such as oil and gas exploration and marine renewable developments [1–3]. This has been driven by concerns for cetaceans, because their behavioural repertoires involve extensive vocalizations and responses to

natural sounds across a broad range of frequencies [4]. To ensure that new developments meet international conservation agreements, modelling frameworks have been developed to explore whether behavioural responses to anthropogenic noise result in population level impacts [5–7]. However, data to parametrize important inputs or components of these models, notably dose–response relationships describing behavioural responses to noise exposure [8], are often sparse or absent.

Opportunities to address these data gaps by tracking behavioural responses of tagged individuals or using arrays of passive acoustic monitoring (PAM) sensors are rare [9–12]. Alternatively, responses may be measured indirectly, using population level changes in occurrence or density in and around exposed areas [13,14]. Behavioural response studies can be further divided into those using an experimental approach and those using an observational approach. Due to the potential link with atypical mass-stranding events, there have been a number of recent experimental dose–response studies on the effects of naval sonar sounds on cetaceans [8,15]. Other studies have determined species specific dose–response relationships for marine mammal behavioural responses to experimental air gun noise (e.g. [16]), routine vessel noise (e.g. [17]) and pile driving [18]. Insights that can be drawn from these studies are constrained by small sample sizes and recognition that the scale of any response may differ between species [8], or in relation to behavioural context [19]. For example, individuals may respond differently to the same stimulus depending upon whether they are foraging or travelling at the time of exposure. Furthermore, most studies of marine mammal responses to anthropogenic noise have examined responses to novel stimuli. Where activities such as seismic surveys or pile-driving extend for periods of weeks or months, the level of response may also change over time as a result of habituation or tolerance [20]. Understanding how these responses change over time is particularly important when predicting potential cumulative impacts of disturbance. However, data are currently lacking on how behavioural responses vary during prolonged periods of noise exposure, constraining attempts to assess overall levels of displacement during large-scale industrial projects.

Harbour porpoises are the most common marine mammal in many areas exposed to offshore energy developments [21,22]. The UK has recently established extensive Special Areas of Conservation for this species under the EU Habitats & Species Directive, and is proposing management measures to avoid significant disturbance from pile-driving noise within these sites [23]. Studies of harbour porpoise displacement in response to pile-driving during wind farm construction have been conducted at a number of North Sea sites, indicating that animals may be disturbed at distances of up to 26 km (e.g. [13,24,25]). These data have provided a conservative estimate of the effective deterrence radius around pile driving activity [26], which is now being used in assessments of the potential significance of displacement within protected areas [23]. While an appropriate first step for the precautionary management of these activities, additional data on the spatial and temporal variation in response levels is urgently required to broaden these assessments. First, most studies of harbour porpoise responses to wind farm construction have been carried out at sites where acoustic deterrent devices (ADD) are also used to mitigate against near-field injury. Given that ADD devices alone may have far-field disturbance effects [27], it remains unclear to what extent observed responses result from pile-driving noise as opposed to other noise sources such as ADDs. Second, while the use of conservative estimates that assume complete displacement within this radius supports precautionary management and mitigation of disturbance, this can constrain broader life-cycle assessments of offshore developments [28]. For example, technologies developed to reduce propagation of piling noise and management measures to reduce simultaneous piling events may reduce disturbance [29,30] but could increase offshore vessel activity and construction timescales. Like the use of ADDs, this may impact harbour porpoises directly through alternative disturbance pathways, or have additional environmental costs through increased energy use and carbon emissions. Better understanding of the scale of any potential disturbance is therefore required to optimize mitigation measures that aim to reduce overall environmental impacts.

In this study, we aimed to inform these policy and management decisions by investigating two key questions. (1) How do harbour porpoise behavioural responses to construction noise vary in relation to: (a) received noise levels; (b) distance from piling; (c) time since the start of construction; and, (d) the duration of individual piling events? (2) To what extent is this response modified by: (a) ADD use prior to piling; and (b) vessel activity? The overall aim was to estimate a proxy porpoise dose–response curve to construction noise in order to refine predictions of the number of individuals displaced by pile-driving.

## 2. Material and methods

Following the approach used to study responses of harbour porpoises to a seismic airgun survey [14], we used echolocation detectors and noise recorders to model harbour porpoise detections along a gradient

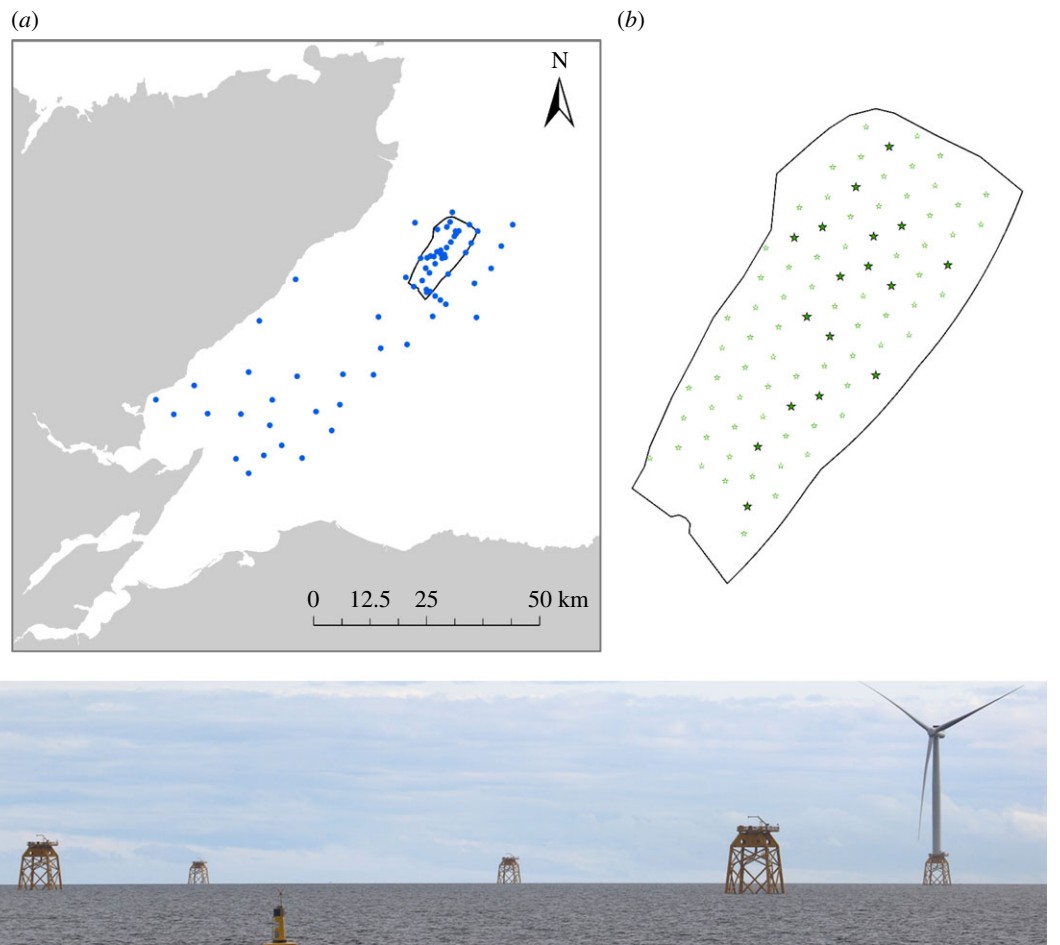

**Figure 1.** The study area: (*a*) showing the location of the BOWL construction site and PAM sampling sites (blue circles); (*b*) detail of the BOWL site showing the 17 piling locations used in the analysis of harbour porpoise responses to construction activity (black-outlined stars). Other turbine sites are shown as smaller green stars. Below is a view of the construction site in August 2018 once steel jackets had been installed at each site and the first turbines were operational.

of exposure to the construction of the Beatrice Offshore Windfarm Ltd (BOWL) (figure 1). Between 2nd April and 2nd December 2017, impulsive pile driving techniques were used to install a set of four 2.2 m diameter steel piles at each of two Offshore Transformer Modules (OTM) and 84 wind turbine locations. The piling vessel was first anchored at each site, and four piles were placed in a pile installation frame that had been lowered onto the seabed. Each pile was then hammered into the sediment using a 1800 or 2400 kJ hammer, with an average piling duration of 5.0 h per set of four piles (range: 2.9–8.8 h).

Underwater noise levels were recorded between March and October 2017 at six locations (figure 1) using autonomous noise recorders (Wildlife Acoustics SM2M Ultrasonic and Ocean Instruments SoundTraps). Recorders were independently calibrated as described in [31]. Measurements were made at a sampling rate of 96 kHz, recording continuously with the SM2Ms and for 10 min per hour with the SoundTraps. Data were analysed in PAMGuide [32] to determine received noise levels. These received levels were used to model piling source levels, taking account of local bathymetry, tide levels and sediment types [33,34]. Further details of this modelling are provided in the electronic supplementary material. Modelled source levels were then used to predict the received single-pulse sound exposure levels (SEL) at all PAM sites (figure 1) for a hammer strike with the maximum hammer energy recorded at each OTM/turbine location (e.g. Figure 2). Predicted SELs were then frequency weighted with three different filter functions to compare responses to broad-band noise levels and those in the frequency ranges most likely to be heard by porpoises. These different functions were: (1) the high-frequency cetacean weighting function proposed by Southall *et al*. [4]; (2) the more recent generalized weighting function for high-frequency cetaceans proposed by NOAA [35,36], and replicated in the updated Southall criteria [37]; and (3) a species specific audiogram (see electronic supplementary material, figure S1) for harbour porpoises [38].

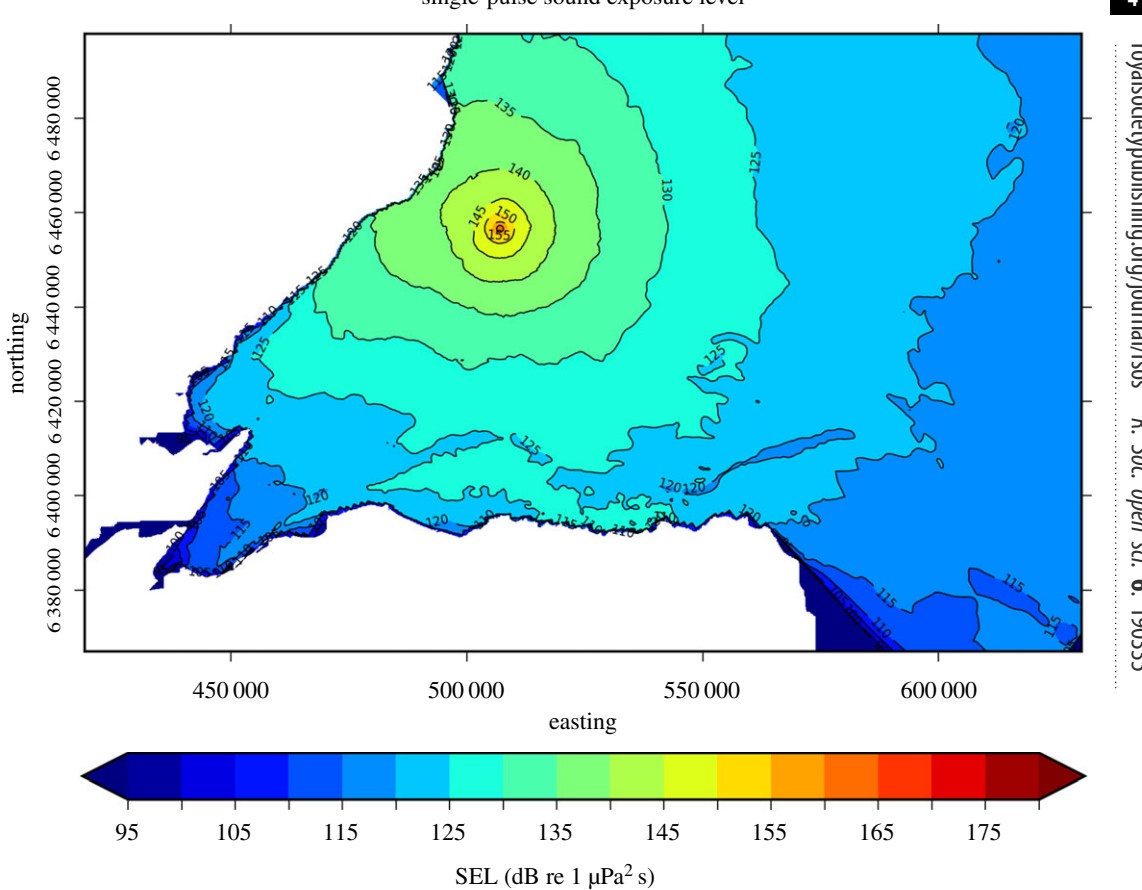

single-pulse sound exposure level

SEL (dB re 1 μPa² s)

**Figure 2.** Modelled predictions of received levels of noise from impact piling at the first location piled, OTM location G7, in the BOWL construction site. Predictions are depth averaged unweighted received single-pulse SEL for a hammer strike of 662 kJ.

Spatio-temporal variation in the occurrence of porpoise echolocation clicks was measured using V.O and V.1 CPODs (www.chelonia.co.uk). We assume that variations in echolocation click detections provide a robust index of changes in the occurrence of porpoises, as indicated by previous work in this study area demonstrating relationships between echolocation detections and two independent measures of relative density derived from visual and digital aerial surveys [39]. CPODs were moored at 68 locations between 0.4 and 76.5 km from turbine locations to provide a gradient of exposure to pile-driving noise (figure 1). Data were successfully recovered from 100 out of 105 deployments between 17 February 2017 and 31 December 2017. Data were downloaded and processed using v. 2.044 of the manufacturer's custom software to identify porpoise echolocation clicks. Click trains categorized as high or moderate quality were used for analyses. Changes in porpoise occurrence (Detection Positive Hours, DPH; [39]) were estimated for each location in a 12- and 24-h period from the end of piling relative to a baseline occurrence of the same duration before the piling event to account for temporal changes in baseline levels of detections that could occur due to underlying seasonal patterns of occurrence [40] or seasonal changes in environmental conditions influencing detection probability [41]. Harbour porpoise detections exhibit diel variation [42] and as the time elapsed from the start to the end of piling at each turbine location was typically less than 12 h, the baseline for the 12-h response was chosen to commence 48 h before the end of piling to ensure that the baseline and response periods were matched with respect to time of day (figure 3). For the 24-h response, as the baseline and response periods covered a full diel cycle, diel variation was not a concern and the baseline was chosen to commence 48 h before the start of piling to avoid overlap of the baseline period with pre-piling activities such as anchoring and placement of the piles into the installation frame (figure 3). To allow sufficient time between piling events for this baseline period, we focused our analysis on responses to pile-driving at 17 turbine locations, where the interval between piling at the previous location and the current location exceeded 96 h (figure 1 inset; electronic supplementary material, table S2).

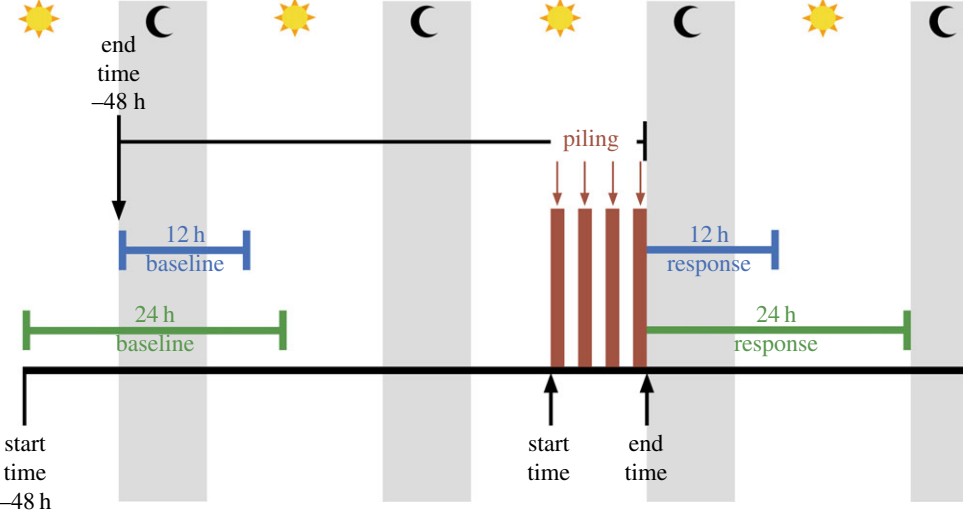

**Figure 3.** Schematic of the timeline for one piling event, indicating the start and end times of the 12-h and 24-h response and baseline periods with respect to the start and end of piling and the diel cycle.

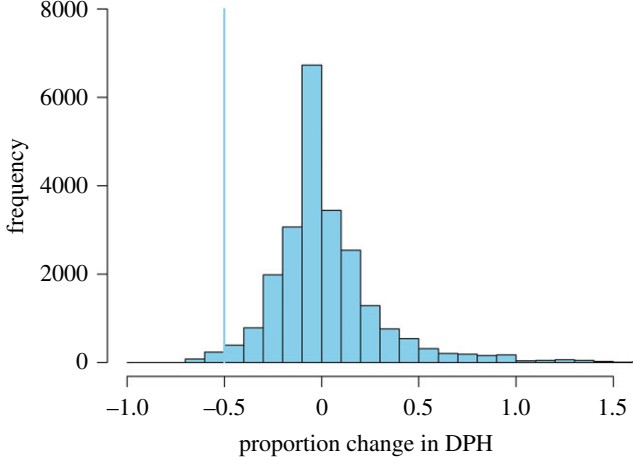

**Figure 4.** Frequency distribution of the proportion change in harbour porpoise occurrence (DPH) for a 24-h period from 1000 randomly sampled times at 12 sites from 07 March 2017 to 16 March 2017 and from 07 December 2017 to 16 December 2017. The blue line indicates the 1st percentile of the distribution.

To characterize baseline variation in day to day changes in occurrence, we used data from 7th to 16th March 2017 before piling began and from a second period from 7th to 16th December 2017 after piling had stopped from 12 similar offshore sites located outside the wind farm site at least 25 km from the construction site. These data were used to produce a null distribution of proportional change in occurrence (DPH) by randomly sampling 1000 times from 9th to 15th March 2017 and from 9th to 15th December 2017 for each site and determining the proportion change in the number of DPH in the 24-h period following each randomly selected time relative to the number of DPH in the 24-h period 2 days prior to it (figure 4). Using the quantile function in R [43], the 1% quantile of this distribution was calculated. Using these data, porpoises were considered to have exhibited a behavioural response to piling when the proportional decrease in occurrence was greater than 0.5, the 1st percentile of the baseline distribution (figure 4). For consistency, the same threshold (0.5) was used for both 24-h and 12-h responses.

The probability that porpoise occurrence did (1) or did not (0) show a response to piling was modelled as a binomial response with a probit link function [17] using generalized linear mixed models (GLMM) in R [43,44]. Distance to piling, on a logarithmic scale, and received single-pulse SEL were used as explanatory variables in separate models because these variables were highly collinear. To examine variation in the response to piling over the eight month construction period, we included the cumulative number of locations piled. Other variables included were ADD use (factor) and the

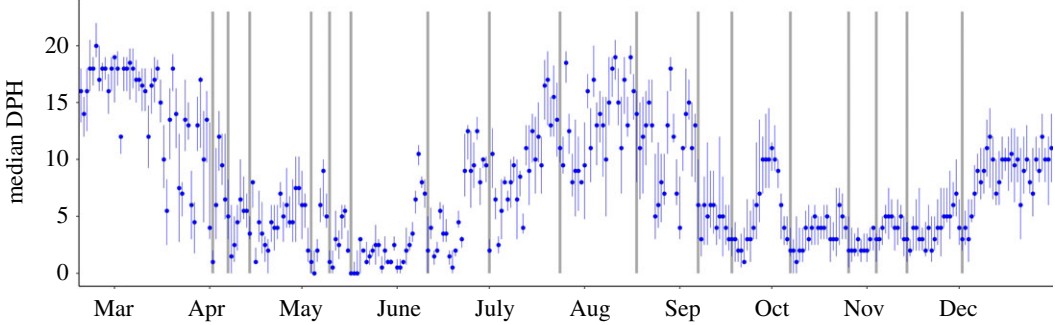

**Figure 5.** Variation in daily porpoise occurrence (median detection positive hours per day $\pm$ interquartile range) on all CPODs in the BOWL construction site February – December 2017. Grey bars indicate the timing of the 17 piling events used in the analysis of responses to piling.

duration of active piling at each turbine site. To control for disturbance by vessel activity, we used Automatic Identification System (AIS) detections obtained at 5-min intervals to estimate the number of vessels within either 1 km or 500 m of each CPOD during the 12-h or 24-h response period [31,45]. Analyses were based on relative changes in click detections from multiple CPOD deployments, therefore site-specific differences, resulting either from differences in individual CPOD sensitivity or site-specific environmental conditions, were accounted for by including a random effect in the model that combined CPOD site and CPOD identity. The acf and pacf functions in R [43] were used to check for autocorrelation in the model residuals and package DHARMa was used for residual diagnostics to validate selected models [46]. Model selection was carried out using Akaike Information Criterion (AIC) [47] and the significance of fixed effects was tested with Likelihood Ratio Tests (LRT) using the anova function in R [43].

## 3. Results

Harbour porpoises were present within the windfarm construction site throughout the construction period in 2017 (figure 5). The number of detection positive hours fluctuated during the year but there was no evidence of a negative temporal trend in occurrence in 2017 as a result of piling.

For both 12-h and 24-h responses, harbour porpoise responses were best explained by the interaction between the cumulative number of locations piled and either distance from piling on a logarithmic scale or audiogram-weighted single-pulse SELs (table 1; electronic supplementary material, table S3). The difference between the two best models of 24-h response was small ($\Delta$AIC = 1.6). In both cases, there was a decrease in response as the number of locations that had been piled increased (figure 6). The same covariates with very similar parameter estimates were retained in a model of 24-h response with distance using a subset of piling events ($n = 9$) preceded by a longer 192-h break in piling (electronic supplementary material, table S4). Based on the relationship with distance from piling, at the start of the construction period in April there was greater than or equal to 50% chance of harbour porpoises responding to piling in the 24-h period after piling at distances up to 7.4 km (95% CI = 5.7–9.4) from piling (figure 6a). By the 47th location in July, this threshold had decreased to 4.0 km (95% CI = 2.7–5.2), declining further to 1.3 km (95% CI = 0.2–2.8) by December, when the final (86th) location was piled. Similarly, there was a greater than or equal to 50% chance of porpoises responding in the 24-h period after piling to audiogram-weighted SEL of 54.1 dB re 1 $\mu$Pa$^2$ s (95% CI = 52.0–56.7) at the first location piled, increasing to 60.0 dB re 1 $\mu$Pa$^2$ s (95% CI = 57.5–63.4) by the 47th location and 70.9 dB re 1 $\mu$Pa$^2$ s (95% CI = 63.0–87.0) by the final location (figure 6b). For the relationship with unweighted single-pulse SEL, there was a greater than or equal to 50% chance of porpoises responding in the 24-h period after piling to unweighted SEL of 144.3 dB re 1 $\mu$Pa$^2$ s (95% CI = 142.1–146.8) at the first location piled, increasing to 150.0 dB re 1 $\mu$Pa$^2$ s (95% CI = 147.5–153.6) by the 47th location and 160.4 dB re 1 $\mu$Pa$^2$ s (95% CI = 153.2–178.9) by the final location (electronic supplementary material, figure S2).

There was no support for including ADD use in models of the 24-h response (LRT test: $\chi_1^2 = 0.708$, $p = 0.40$), but ADD use was a significant covariate in models of the 12-h response (LRT test: $\chi_1^2 = 12.892$, $p < 0.001$; table 1; electronic supplementary material, table S3). Repeating the analysis for 18 turbine locations, including a second location that was piled without ADD mitigation following a

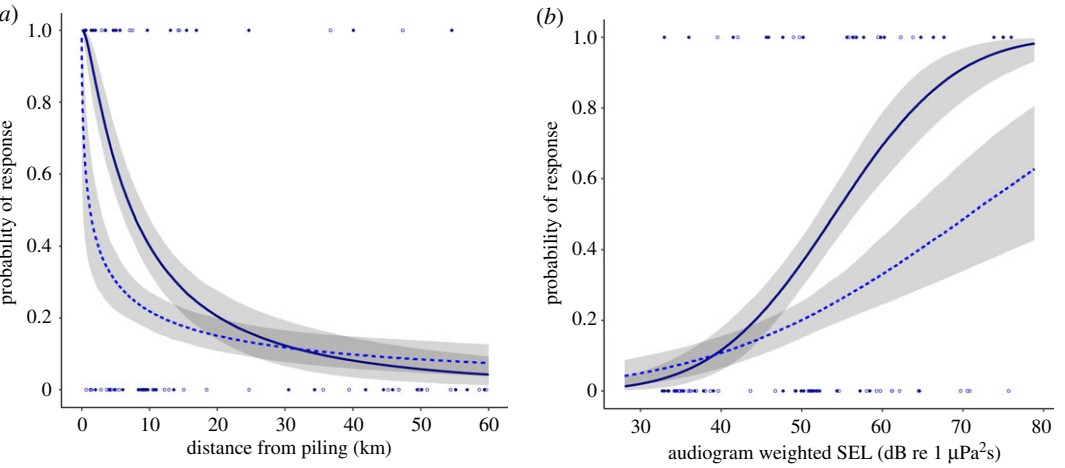

**Figure 6.** The probability of a harbour porpoise response (24 h) in relation to the partial contribution of (*a*) distance from piling and (*b*) audiogram-weighted received single-pulse SEL for the first location piled (solid navy line) and the final location piled (dashed blue line), predicted assuming the number of AIS vessel locations within 1 km = 0; confidence intervals (shaded areas) estimated for uncertainty in fixed effects only. Harbour porpoise occurrence was considered to have responded to piling when the proportional decrease in occurrence (DPH) exceeded a threshold of 0.5. Points show actual response data for the first location piled (filled navy circles) and the final location piled (open blue circles).

**Table 1.** Modelled relationships of harbour porpoise behavioural response to piling. Response was defined as a proportional decrease in harbour porpoise occurrence greater than 0.5 in the 12 or 24 h after cessation of piling. Relationships were modelled using GLMM with a binomial error distribution and the probit link function. Distance from piling, audiogram-weighted received single-pulse sound exposure levels (ASS_SEL), cumulative number of locations piled (piling order), ADD use and the number of AIS vessel locations within either 500 m or 1 km were used as explanatory variables. All models included a random effect of CPOD sampling site combined with CPOD identity: model (*a*) variance = 0.027, s.d. = 0.165; (*b*) variance = 0.022, s.d. = 0.149; (*c*) variance = 0.159, s.d. = 0.398.

| model | estimate | s.e. | z-value | p-value | AIC |
|---|---|---|---|---|---|
| (*a*) 24-h response ∼ log(distance) * piling order + no. vessel locations_1 km | | | | | 619.4 |
| (intercept) | 0.8352 | 0.1548 | 5.397 | <0.001 | |
| log(distance):piling order | 0.1864 | 0.0597 | 3.123 | 0.002 | |
| log(distance) | −0.5734 | 0.0616 | −9.305 | <0.001 | |
| piling order | −0.6431 | 0.1539 | −4.178 | <0.001 | |
| no. vessel locations_1 km | 0.2025 | 0.0945 | 2.143 | 0.032 | |
| (*b*) 24-h response ∼ ASS_SEL * piling order + no. vessel locations_1 km | | | | | 621.0 |
| (intercept) | −0.6798 | 0.0667 | −10.188 | <0.001 | |
| ASS_SEL:piling order | −0.2088 | 0.0711 | −2.938 | 0.003 | |
| ASS_SEL | 0.6857 | 0.0734 | 9.342 | <0.001 | |
| piling order | −0.1624 | 0.0625 | −2.598 | 0.009 | |
| no. vessel locations_1 km | 0.2118 | 0.0945 | 2.240 | 0.025 | |
| (*c*) 12-h response ∼ log(distance) * piling order + ADD + no. vessel locations_500 m | | | | | 653.4 |
| (intercept) | 0.2079 | 0.3202 | 0.649 | 0.52 | |
| log(distance):piling order | 0.2641 | 0.0673 | 3.922 | <0.001 | |
| log(distance) | −0.5844 | 0.0745 | −7.843 | <0.001 | |
| piling order | −0.6777 | 0.1811 | −3.742 | <0.001 | |
| ADD | 0.9381 | 0.2849 | 3.292 | <0.001 | |
| no. vessel locations_500 m | 0.6042 | 0.4443 | 1.360 | 0.17 | |

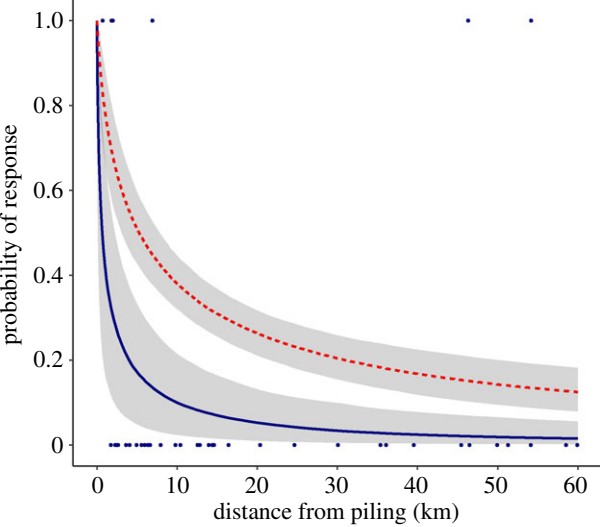

**Figure 7.** The probability of a harbour porpoise response (12 h) in relation to the partial contribution of distance from piling with (dashed red line) and without (solid navy line) the use of the ADD prior to piling, predicted for the 62nd and 61st location piled, respectively, assuming the number of AIS vessel locations within 500 m = 0; confidence intervals (shaded areas) estimated for uncertainty in fixed effects only. Harbour porpoise occurrence was considered to have responded to piling when the proportional decrease in occurrence (DPH) exceeded a threshold of 0.5. Points show actual response data for the 61st location piled (filled navy circles), which was piled without the use of the ADD prior to piling.

break in piling of 3.7 days, showed that ADD use remained a significant covariate (LRT test: $\chi_1^2 = 17.889$, $p < 0.001$; electronic supplementary material, table S3) and did not change the model. Figure 7 shows the predicted responses to piling with and without ADD use for the 62nd and 61st location piled, respectively: the 61st location was the location included in the analysis that was piled without ADD mitigation (electronic supplementary material, table S2). The response to piling and ADD use was greater than the response to piling alone, with a greater than or equal to 50% chance of harbour porpoises responding in the 12-h period after piling at distances up to 5.3 km (95% CI = 3.1–7.8) from piling with prior ADD use but only up to 0.7 km (95% CI = 0.1–2.3) from piling without ADD use (figure 7), at the 62nd and 61st location, respectively.

All the best models included a covariate of vessel numbers to control for vessel activity within the proximity of the CPOD. For models of the 24-h response, this was the number of AIS vessel locations within 1 km of the CPOD, whereas for models of the 12-h response this was the number of AIS vessel locations within 500 m of the CPOD (table 1). In all cases, higher vessel activity increased the probability of observing a response, which could indicate either a response of porpoises to vessels, a masking of porpoise detections on the CPOD by vessel noise, or both. Audiogram-weighted single-pulse SEL was a better predictor of harbour porpoise responses than NOAA weighted [35], M weighted [4] or unweighted single-pulse SEL (electronic supplementary material, table S3).

# 4. Discussion

Our results provide a behavioural response curve that relates the proportion of the local porpoise population disturbed to distance from piling, which can now be used to improve estimates of the number of individuals disturbed in population level assessments of the impacts of windfarm construction [5,6]. Furthermore, we found that the scale of response by the local population of porpoises declined over time, highlighting that previous assessments of disturbance impacts of long-term piling programmes may be conservative [24]. Despite smaller sample sizes, there was preliminary evidence that shorter-term responses to the cumulative impact of ADD and impact piling were greater than responses to pile driving alone. Similarly, higher vessel activity was associated with an increased probability of response, and the porpoises' response to noise was best explained by distance to piling or received noise levels within their high-frequency hearing range. Together, these findings suggest that management efforts to reduce exposure to low-frequency impulsive piling noise

should be carefully balanced against potential disturbance by other noise sources associated with construction and impact mitigation.

## 4.1. Methodological limitations

Dose–response relationships can only be determined from individual-based studies [8]. However, even on the rare occasions where this has been achieved through tagging studies of large whales, inferences are often constrained by small sample sizes [8,15]. The logistical challenges involved in such work are even greater for small cetaceans, and harbour porpoises have only been tagged routinely in areas where they are bycaught in fisheries [48]. While acoustic arrays can be used to track individuals of some species, high-frequency cetaceans such as harbour porpoises can only be tracked over small-scale arrays [49]. Consequently, the majority of studies aimed at understanding disturbance to this widespread and abundant species have been carried out using dispersed arrays of independent PAM sensors or visual aerial surveys [12,14,24,29,40]. These PAM studies only measure population responses, and our results (e.g. figure 6) therefore provide only a proxy for a dose–response curve. As such, while they can be used to estimate displacement or habitat loss, they represent the integration of many individual responses, which may vary as a result of individual differences in sensitivity or variation in behavioural context [19]. While we cannot identify the behavioural state of individuals exposed, it seems reasonable to assume that the distribution of contexts across the population, over multiple piling events, is representative of similar North Sea habitats. These proxy dose–response functions should therefore be applicable to similar offshore windfarm development sites. Because we were unable to follow individuals, we cannot determine whether or not the decline in response during construction resulted from habituation, we can only say that harbour porpoises in the construction site showed a smaller response to pile driving noise at the end of the construction period than at the beginning [20]. It is also unknown whether a similar sample of the population were present in the study area during different stages of construction. For example, it is conceivable that more sensitive individuals that fled early in the season could have been replaced by new individuals that were less responsive [8,50]. These uncertainties highlight the need to consider prior conditions and cetacean residence patterns in any proposed development area. This was the first commercial windfarm in our study area, but prior to this the population had experienced decades of oil and gas exploration [14,51]. Consequently, one should be cautious about using these findings in less industrial areas where porpoises have not previously been exposed to impulsive noise.

One limitation of our approach to estimating behavioural response functions is that we required a suitable baseline period prior to the piling activity, which could then be compared to a reference period once piling had ceased. This restricted our analysis to a subset of only 17 (20%) piling events during construction, when operational or weather conditions had caused delays that provided a suitable baseline period. This also meant that we were unable to examine finer-scale or instantaneous responses. Consequently, the results represent a response to cumulative exposure to pile driving, vessel activities and the use of ADD. Part of the rationale for focusing our analysis on these baseline and reference periods was to minimize potential effects of poor signal-to-noise ratio on detection probability during periods of piling. Porpoise click detections on CPODs decrease with increasing noise around oil and gas platforms, particularly between 20 and 160 kHz [52]. By starting the response period at the end of pile driving, we avoided the period when noise might have had the greatest effect on CPOD detections. Noise from construction vessels may still have affected CPOD detection probability close to piling locations for a few hours after piling ceased. However, removing data from all locations within 1 km of piling locations did not change our results (see electronic supplementary material, table S3), suggesting that our analyses were robust to background noise issues. Nevertheless, the effect of different sources of noise on CPOD detection probability requires further investigation to optimize the design of studies which might disentangle the role of different noise sources in shaping observed responses.

Another consideration is whether decreases in detections resulted from changes in vocalization rates rather than displacement. One of two tagged harbour porpoises exposed to close vessel passes ceased echolocating for several minutes [10], therefore, it is possible that harbour porpoises ceased vocalizing in response to pile-driving, although it seems less likely that a species with such high vocalization rates [53] would cease vocalizing for several hours. Previous studies that detected reductions in CPOD detections in response to seismic surveys [14] and pile driving [13] provided additional evidence of displacement through parallel aerial surveys. We suggest that displacement is the main driver of observed changes in echolocation detections in our study. However, even if this is not the case,

prolonged cessation of vocalization is as much a response as displacement and is considered to be a similar severity of behavioural disturbance to a moderate shift in group distribution [4]. Indeed, the impact of remaining in the disturbed area at the expense of vocalizing and feeding, could even be larger than rapidly fleeing the area and resuming feeding [54].

Finally, detection probability will be affected by a number of environmental variables including depth, temperature, salinity and bottom substrate. In the Baltic, Carlén *et al*. [41] estimated spatial and temporal variation in the effective detection area for CPODs and found that probability of detection varied in relation to porpoise density, region and month of the year. It is likely that detection probability also varied spatially and temporally in our study. However, the use of the proportional change in porpoise detections, with baseline and response periods matched spatially and temporally, should have accounted for much of the variation in detection probability. The threshold proportional change in occurrence that we used to define a response was based on data from immediately before and after the 10-month piling period, and we assume that this was representative of the whole study period. To be consistent, we used the same threshold for both 12-h and 24-h responses. Given the 1st percentile of the distribution of baseline proportional change in occurrence for the 12-h response was slightly lower ($-0.67$) than the 24-h value ($-0.5$), the application of a consistent $-0.5$ threshold means that results based upon the 12-h response will be slightly more precautionary.

## 4.2. Management implications

Given the widespread distribution and abundance of harbour porpoises in the North Sea, potential disturbance impacts on protected populations must be considered within consent applications for most, if not all, wind farm developments in this region. Several modelling frameworks now exist for predicting the population effects of such disturbance [5–7], but there has been uncertainty over the spatial scale of responses to piling noise and how these change over time. The interim Population Consequences of Disturbance (iPCoD) model [5] is widely used by UK developers and regulators to model potential population consequences of alternative construction scenarios. Our results can now be used to improve estimates of the number of individuals disturbed during piling events; one of the model's key input parameters. These estimates require underlying density distributions for the development area of interest, typically based upon data collected through broad-scale international surveys [22,55]. Previously, density data have been used in conjunction with threshold distances (table 1 in [56]) or modelled noise levels [2] to estimate numbers of individuals disturbed. For example, current guidance for UK Harbour Porpoise SACs suggests that complete displacement should be assumed over a 26 km radius around pile-driving [23]. Our results indicate that this approach is highly conservative. Based upon an average density of porpoises in our Moray Firth study area of $0.274\ km^{-2}$ [22], the JNCC guidance predicts displacement of 582 individuals. In comparison, 160 (95% CI = 120–202) and 102 (95% CI = 75–133) individuals are predicted to be disturbed based upon our behavioural response function for the first and last piling events, respectively, 28% (95% CI = 21–35) and 18% (95% CI = 13–23) of the total estimate of 582 individuals if using current guidance (electronic supplementary material).

Policy instruments in many countries aim to reduce anthropogenic noise that may adversely affect the marine environment [57]. However, uncertainty over the relative risk of different noise sources and their pathways to impact has constrained efforts to translate these aims into management practice. Impulsive noise sources such as pile driving have been a focus of concern for many stakeholders, and conservative estimates of responses to piling may affect previous estimates of the benefits of wide-scale use of noise reduction technologies [58]. A recent process-based model for assessing the impacts of windfarm construction on North Sea harbour porpoise populations was most sensitive to the distance at which animals responded to pile driving noise, and population effects were only evident when response distances exceeded 20–50 km [6]. Data from our study suggest that response distances are unlikely to exceed 20 km, and provide a dataset that can be incorporated into available population modelling frameworks to undertake more detailed cost–benefit analyses of potential noise reduction methodologies.

Efforts to reduce behavioural disturbance must also be balanced against efforts to reduce the risk of near-field injuries as a result of loud impulsive pile-driving noise. In Germany, it is mandatory to deploy an ADD at least 30 min before piling to mitigate the risk of physical injury. Consequently, most previous observational studies of harbour porpoise responses to pile driving noise, represent responses to the cumulative impact of ADDs, pile driving and associated construction vessels. For example, ADDs were used during the construction of all the commercial windfarms in the studies cited in table 1 in [56]. ADDs were also used during most piling events in our study, albeit for a shorter 15 min period

prior to piling. Thus, the 24-h behavioural response function (figure 6) represents a response to the cumulative impact of construction vessels, ADD and the pile driving itself. The 12-h response function however includes a covariate of ADD use, allowing the partial response to construction vessels and pile driving noise to be estimated in the absence of ADD mitigation. There was only one location where a 4-day break in piling was followed by installation without ADD mitigation, giving us limited power to disentangle the responses to the different noise sources. Nevertheless, our results suggest that responses were increased by the use of ADDs prior to piling, and although there was no replication of the stimulus (i.e. piling without ADD exposure), there were 47 CPODs measuring the response to that stimulus. Although these were only pseudo-replicates in relation to the stimulus, they do represent true replicates with respect to the response. We were also able to repeat the analysis using a second location that was piled without ADD mitigation following a break in piling of 3.7 days, demonstrating that ADD use remained a significant covariate (see electronic supplementary material, table S3). As a true replicate in relation to the stimulus, this adds more weight to the argument that measured responses were a result of the absence of ADD mitigation and not caused by some other confounding factor on a single day. The contribution of ADDs to behavioural responses to windfarm construction have often been overlooked, but results of other studies of ADD use [27,29] highlight their potential contribution to these responses. Policy and management guidance should consider how best to balance these different sources of disturbance during construction, but this requires further exploration of the consequences of any trade-off between using ADDs to reduce near-field risk of injury and minimizing far-field disturbance. Vessel presence within 1 km was also a significant covariate in our models, possibly indicating a near-field behavioural response of porpoises to vessels that could potentially contribute significantly to the cumulative impact of the construction phase as a whole. Alternatively, the noise from vessels in close proximity to CPODs could have masked porpoise detections [52]. Previous studies of cetaceans have studied interactions with vessels in other contexts, but further work is required to better understand the relative contribution of pile driving, ADD noise and vessel activity to observed responses of cetaceans to offshore construction.

Policy and management measures aimed at minimizing the environmental impacts of wind farm construction on marine mammals have tended to focus on high-energy pile-driving noise. These impulse noise sources are a more significant risk with respect to near-field injuries, but there is more uncertainty over how noise influences behavioural reactions. If animals are responding to a perceived threat, then their reactions may be more closely related to an individual's distance to the source rather than received noise levels. Either way, their perception of loudness or other signal characteristics will depend on the shape of their audiogram [59] leading to recommendations for the use of audiogram-weighted noise metrics when assessing impacts [19,56]. At one level, our results support this approach, as audiogram-weighted SELs were marginally better predictors of behavioural response than either unweighted, high-frequency cetacean weighted or NOAA weighted noise levels. However, as distance to piling and all weighted and unweighted received noise levels were highly correlated, our study was not well suited to testing which was a better predictor of response. Distance to piling was as good a predictor as the weighted noise level estimates (electronic supplementary material, table S3), and the use of distance rather than modelled audiogram-weighted received levels would be a much more pragmatic and transparent response variable for large-scale assessments. In future, integration of data from multiple windfarm sites could provide a wider range of received noise levels at given distances from piling, helping disentangle the relative importance of distance to source and loudness in shaping behavioural responses. Comparative studies across sites with contrasting mitigation procedures and installation fleets are now also required to understand how to minimize overall levels of disturbance during both construction and operation of offshore windfarms.

Ethics. This was a non-invasive, acoustic observational study of harbour porpoise responses to pile-driving. The authors had no control or influence over the duration or scheduling of pile-driving, which occurred during development of a commercial offshore windfarm. No animals were captured or tagged during this study and no research or animal ethical assessments were required. Porpoise responses were determined using remote passive acoustic devices on seabed moorings licensed for scientific use by Marine Scotland, and consented by the Crown Estate. Moorings were deployed and recovered using vessels with appropriate certification, accreditation and endorsements.
Data accessibility. The article's data and R code are available from the Dryad Digital Repository: https://doi.org/10.5061/dryad.5qg30sd [60].
Authors' contributions. I.G. and P.T. conceived and designed the study. T.B. managed the data collection. I.G., B.C. and S.B. processed the data. N.M. and A.F. analysed all the noise data. I.G. carried out the data analysis. I.G. and P.T. drafted the manuscript, which all authors subsequently reviewed and edited.

Competing interests. The authors have no known competing interests. This study was funded by a commercial developer, Beatrice Offshore Wind Ltd. (BOWL). However, the funding body had no input into data collection, data analysis or interpretation, or the writing of the paper. The aims, scope and experimental design of the study were developed by the authors to meet BOWL planning consent conditions. These were agreed by the regulator Marine Scotland Licensing and Operations Team following consultation with statutory advisors represented on the Moray Firth Regional Advisory Group (MFRAG); a stakeholder group that was established by the Scottish Government to oversee the monitoring programme.

Funding. This study was funded by Beatrice Offshore Wind Ltd. (BOWL) using equipment previously purchased by UK Department of Energy & Climate Change, Scottish Government, Oil and Gas UK, COWRIE and Moray Offshore Renewables Ltd. P.T. and T.B. were core funded by University of Aberdeen. N.M. and A.F. were core funded by Centre for Environment, Fisheries and Aquaculture Science. I.G. and B.C. were core funded by University of Aberdeen but with salary support for the period of this study though contract to BOWL. S.B. was a self-funded post-graduate student.

Acknowledgements. We thank Bill Ruck and colleagues from the University of Aberdeen and Moray First Marine for support and assistance with data collection. We also thank Nick Brockie, Lis Royle, Elizabeth Reynolds and the rest of the team at BOWL for details of the construction programme and for facilitating data collection during construction of the windfarm, and members of MFRAG for their constructive advice.

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
