## [Reviewer comments · Royal Society Open Science]

Review History

RSOS-190335.R0 (Original submission)

Review form: Reviewer 1

Is the manuscript scientifically sound in its present form?

Yes

Are the interpretations and conclusions justified by the results?

Yes

Is the language acceptable?

Yes

Is it clear how to access all supporting data?

Yes

Do you have any ethical concerns with this paper?

No

Have you any concerns about statistical analyses in this paper?

Yes

Recommendation?

Accept with minor revision (please list in comments)

Comments to the Author(s)

General comments

In this interesting and timely study, Graham and colleagues investigate harbour porpoise responses to pile driving over a period of several months, and explore the factors that may contribute to the likelihood of responding. Results are important for the management of these developments and for devising appropriate strategies to minimize impacts during construction. The sampling design is robust, and results are presented clearly and concisely. I particularly appreciated the extensive discussion of the practical implications of these findings. I have a couple of general comments, and a few more detailed ones that I list below.

Throughout the manuscript, I found that indications of the uncertainty in model results and predictions were lacking. For example, the distances at which the expected response is 50% or the percentages of individuals displaced within a 26-km radius are presented as single numbers. This carries the danger of these numbers being used as fixed truths in future policies (just as the 26 km radius is used now). Rather than reporting single expected values, I would refer to the confidence intervals.

I found the description of the approach used to define a response slightly confusing (as I detail below). Overall, I like the solution the authors have devised, but it took a while to wrap my head around it, so I suggest rethinking the presentation of the methods to improve clarity (below I suggest using a sample timeline with the indication of the various periods).

Finally, I have some concerns on the use of proportional changes in occurrence as a measure of response. This should in theory account for any seasonal variability in occurrence, but, if occurrence is very low in some periods of the year, proportions likely become noisier, and a temporal trend in the response becomes harder to tease apart from the temporal trend in occurrence (see also my comment below on the short reference window used). Another complication of using proportional change is that it cannot disentangle permanent changes in occurrence; in other words, it is currently assumed that a 96-h interval is sufficient for the occurrence to return to baseline, but porpoises could be leaving and not returning. Remaining individuals may be less responsive (as the authors briefly discuss) and more responsive individuals could be simply kept away from these locations because the interval between events is not sufficient for them to return. This would therefore not be an indication of diminishing response or tolerance/habituation (as implied in most of the manuscript, including the title), but rather of an intense, prolonged effect. I think it is worth checking whether there is any indication of these processes occurring, for example by repeating the analysis only using events separated by longer gaps (192 h?), or investigating any temporal trend in occurrence in baseline periods (and comparing this trend with the trend in responses). These checks could be included in the Supplementary material.

Detailed comments

Line 16: I would clarify that 'location' refers to the location of piles being driven; this only became clear after reading the Methods section.

Line 37: Please cite the recent review of the population consequences of disturbance framework (Pirootta et al. 2018, *Ecology and Evolution*, <https://doi.org/10.1002/ece3.4458>).

Line 42: You could cite the work by Miller et al. 2014 (*JASA*, doi: 10.1121/1.4861346) on killer whale dose-response analysis. Also, it is worth mentioning the study by Moretti et al. 2014 (*PLoS One*; doi: 10.1371/journal.pone.0085064) in this section; they have also used passive acoustics to quantify group-level risk functions.

Lines 85-90: You currently do not list your assessment of the effects of shipping traffic among the aims, which I find as important as the ADD analysis.

Lines 113-116: You may be aware that updated weighting functions have recently been published by Southall et al. (2019; *Aquatic Mammals*, doi: 10.1578/AM.45.2.2019.125). I don't think it's necessary to rerun the entire analysis, but it would be interesting to compare how the weighting differs in the updated version.

Lines 126-129: A few aspects are unclear in this section: 1) How did you compute DPH (which you define as detection positive hours per day) over a 12-h period? 2) Why is the 48-h baseline period calculated from the start of piling for 24 h analysis, and from the end of piling for the 12 h analysis? This seems like an important detail but I struggle to see a justification. 3) Do I understand correctly that a 12-h period is used as a baseline for the 12-h analysis, and a 24-h period for the 24-h analysis? In general, it might help to have a schematic representation of the timeline for one piling event, identifying the two response and the two baseline periods.

Line 135-136: The description of how this baseline variation was computed and used only became clear after reading the Supplementary material. I would consider bringing that section (Figure S1) into the main text, for clarity. Also, I understand your rationale, but this approach implicitly assumes that the proportional change in occurrence in those two weeks in March is representative of proportional changes over an entire year. This could be true, but I could also imagine a scenario where the proportional change in occurrence varies seasonally (if, for instance, porpoises use the area more transiently in one season and more consistently in another). I think this assumption is quite important, and should be mentioned in the Discussion. Also, the baseline proportional change is only calculated for a 24-h period. I'd be curious to know if the 99th percentile change remains the same when considering 12-h windows.

Line 146: From Table S1 and the Discussion, I gather there was only one event where no ADD was used. I agree this is still an important piece of information, and it should be reported. However, with one sample, you cannot rule out the option that some other confounding factor has contributed to porpoises reacting less on that day. I think you should discuss this carefully and tone down your statements regarding the use of ADDs throughout the manuscript. This is an important preliminary observation, but at the moment it is highly correlative.

Line 149: I assume that by 'response period', you mean the 12-h and 24-h windows, and not the period of piling, correct?

Line 152: You should probably test for any remaining autocorrelation in model residuals.

Line 154: Did you use REML to fit your mixed-effects models in lme4? If so, you cannot use AIC to compare models with different fixed-effects components (see here, for example: <https://stats.stackexchange.com/questions/131272/lme4-why-is-aic-no-longer-displayed-when-using-reml>). The models must be refitted using ML for comparison between different fixed-effect

structures (the function extractAIC should do that automatically). In my experience, results often do not change much.

Line 175: 'Highly significant' implies the use of some hypothesis test, of which I could not find any mention in the text. Also, if this is indeed statistical significance, I would remove the word 'highly': after choosing a significance level, a result is either significant or not.

Line 216: Again, I suggest citing Moretti et al. 2014 here.

Line 269: And I would cite Pirotta et al. 2018 here.

Review form: Reviewer 2 (Jakob Tougaard)

Is the manuscript scientifically sound in its present form?

Yes

Are the interpretations and conclusions justified by the results?

Yes

Is the language acceptable?

Yes

Is it clear how to access all supporting data?

Yes

Do you have any ethical concerns with this paper?

No

Have you any concerns about statistical analyses in this paper?

No

Recommendation?

Accept with minor revision (please list in comments)

Comments to the Author(s)

See attached file (Appendix A).

Decision letter (RSOS-190335.R0)

01-Apr-2019

Dear Dr Graham,

The editors assigned to your paper ("Harbour porpoise responses to pile-driving diminish over time") have now received comments from reviewers. We would like you to revise your paper in accordance with the referee and Associate Editor suggestions which can be found below (not

including confidential reports to the Editor). Please note this decision does not guarantee eventual acceptance.

Please submit a copy of your revised paper before 24-Apr-2019. Please note that the revision deadline will expire at 00.00am on this date. If we do not hear from you within this time then it will be assumed that the paper has been withdrawn. In exceptional circumstances, extensions may be possible if agreed with the Editorial Office in advance. We do not allow multiple rounds of revision so we urge you to make every effort to fully address all of the comments at this stage. If deemed necessary by the Editors, your manuscript will be sent back to one or more of the original reviewers for assessment. If the original reviewers are not available, we may invite new reviewers.

- Data accessibility

<http://datadryad.org/submit?journalID=RSOS&manu=RSOS-190335>

- Competing interests

- Authors' contributions

- Acknowledgements

- Funding statement

on behalf of Professor Kevin Padian (Subject Editor)
openscience@royalsociety.org

Subject Editor's comments:

We are prepared tentatively to accept your manuscript but both reviewers raised sufficient concerns that we think a rewrite will help. Please observe our timeline and return your revision, addressing all comments. Thanks for submitting.

Comments to Author:

Reviewers' Comments to Author:

Reviewer: 1

Comments to the Author(s)

General comments

In this interesting and timely study, Graham and colleagues investigate harbour porpoise responses to pile driving over a period of several months, and explore the factors that may contribute to the likelihood of responding. Results are important for the management of these

developments and for devising appropriate strategies to minimize impacts during construction. The sampling design is robust, and results are presented clearly and concisely. I particularly appreciated the extensive discussion of the practical implications of these findings. I have a couple of general comments, and a few more detailed ones that I list below.

Throughout the manuscript, I found that indications of the uncertainty in model results and predictions were lacking. For example, the distances at which the expected response is 50% or the percentages of individuals displaced within a 26-km radius are presented as single numbers. This carries the danger of these numbers being used as fixed truths in future policies (just as the 26 km radius is used now). Rather than reporting single expected values, I would refer to the confidence intervals.

I found the description of the approach used to define a response slightly confusing (as I detail below). Overall, I like the solution the authors have devised, but it took a while to wrap my head around it, so I suggest rethinking the presentation of the methods to improve clarity (below I suggest using a sample timeline with the indication of the various periods).

Finally, I have some concerns on the use of proportional changes in occurrence as a measure of response. This should in theory account for any seasonal variability in occurrence, but, if occurrence is very low in some periods of the year, proportions likely become noisier, and a temporal trend in the response becomes harder to tease apart from the temporal trend in occurrence (see also my comment below on the short reference window used). Another complication of using proportional change is that it cannot disentangle permanent changes in occurrence; in other words, it is currently assumed that a 96-h interval is sufficient for the occurrence to return to baseline, but porpoises could be leaving and not returning. Remaining individuals may be less responsive (as the authors briefly discuss) and more responsive individuals could be simply kept away from these locations because the interval between events is not sufficient for them to return. This would therefore not be an indication of diminishing response or tolerance/habituation (as implied in most of the manuscript, including the title), but rather of an intense, prolonged effect. I think it is worth checking whether there is any indication of these processes occurring, for example by repeating the analysis only using events separated by longer gaps (192 h?), or investigating any temporal trend in occurrence in baseline periods (and comparing this trend with the trend in responses). These checks could be included in the Supplementary material.

Detailed comments

Line 16: I would clarify that 'location' refers to the location of piles being driven; this only became clear after reading the Methods section.

Line 37: Please cite the recent review of the population consequences of disturbance framework (Pirodda et al. 2018, *Ecology and Evolution*, <https://doi.org/10.1002/ece3.4458>).

Line 42: You could cite the work by Miller et al. 2014 (*JASA*, doi: 10.1121/1.4861346) on killer whale dose-response analysis. Also, it is worth mentioning the study by Moretti et al. 2014 (*PLoS One*; doi: 10.1371/journal.pone.0085064) in this section; they have also used passive acoustics to quantify group-level risk functions.

Lines 85-90: You currently do not list your assessment of the effects of shipping traffic among the aims, which I find as important as the ADD analysis.

Lines 113-116: You may be aware that updated weighting functions have recently been published

by Southall et al. (2019; Aquatic Mammals, doi: 10.1578/AM.45.2.2019.125). I don't think it's necessary to rerun the entire analysis, but it would be interesting to compare how the weighting differs in the updated version.

Lines 126-129: A few aspects are unclear in this section: 1) How did you compute DPH (which you define as detection positive hours per day) over a 12-h period? 2) Why is the 48-h baseline period calculated from the start of piling for 24 h analysis, and from the end of piling for the 12 h analysis? This seems like an important detail but I struggle to see a justification. 3) Do I understand correctly that a 12-h period is used as a baseline for the 12-h analysis, and a 24-h period for the 24-h analysis? In general, it might help to have a schematic representation of the timeline for one piling event, identifying the two response and the two baseline periods.

Line 135-136: The description of how this baseline variation was computed and used only became clear after reading the Supplementary material. I would consider bringing that section (Figure S1) into the main text, for clarity. Also, I understand your rationale, but this approach implicitly assumes that the proportional change in occurrence in those two weeks in March is representative of proportional changes over an entire year. This could be true, but I could also imagine a scenario where the proportional change in occurrence varies seasonally (if, for instance, porpoises use the area more transiently in one season and more consistently in another). I think this assumption is quite important, and should be mentioned in the Discussion. Also, the baseline proportional change is only calculated for a 24-h period. I'd be curious to know if the 99th percentile change remains the same when considering 12-h windows.

Line 146: From Table S1 and the Discussion, I gather there was only one event where no ADD was used. I agree this is still an important piece of information, and it should be reported. However, with one sample, you cannot rule out the option that some other confounding factor has contributed to porpoises reacting less on that day. I think you should discuss this carefully and tone down your statements regarding the use of ADDs throughout the manuscript. This is an important preliminary observation, but at the moment it is highly correlative.

Line 149: I assume that by 'response period', you mean the 12-h and 24-h windows, and not the period of piling, correct?

Line 152: You should probably test for any remaining autocorrelation in model residuals.

Line 154: Did you use REML to fit your mixed-effects models in lme4? If so, you cannot use AIC to compare models with different fixed-effects components (see here, for example: <https://stats.stackexchange.com/questions/131272/lme4-why-is-aic-no-longer-displayed-when-using-reml>). The models must be refitted using ML for comparison between different fixed-effect structures (the function `extractAIC` should do that automatically). In my experience, results often do not change much.

Line 175: 'Highly significant' implies the use of some hypothesis test, of which I could not find any mention in the text. Also, if this is indeed statistical significance, I would remove the word 'highly': after choosing a significance level, a result is either significant or not.

Line 216: Again, I suggest citing Moretti et al. 2014 here.

Line 269: And I would cite Pirotta et al. 2018 here.

Reviewer: 2

Comments to the Author(s)
See attached file

Author's Response to Decision Letter for (RSOS-190335.R0)

See Appendix B.

RSOS-190335.R1 (Revision)

Review form: Reviewer 1

Is the manuscript scientifically sound in its present form?

Yes

Are the interpretations and conclusions justified by the results?

Yes

Is the language acceptable?

Yes

Is it clear how to access all supporting data?

Yes

Do you have any ethical concerns with this paper?

No

Have you any concerns about statistical analyses in this paper?

No

Recommendation?

Accept as is

Comments to the Author(s)

I thank the authors for considering and addressing my comments in detail. I am fully satisfied with their responses to my and Reviewer 2's points, and have no further comment. Congratulations on an excellent study.

Decision letter (RSOS-190335.R1)

28-May-2019

Dear Dr Graham,

I am pleased to inform you that your manuscript entitled "Harbour porpoise responses to pile-driving diminish over time" is now accepted for publication in Royal Society Open Science.

on behalf of Prof Kevin Padian (Subject Editor)
openscience@royalsociety.org

Associate Editor Comments to Author:
The paper appears ready for acceptance - congratulations!

Reviewer comments to Author:
Reviewer: 1

Comments to the Author(s)
I thank the authors for considering and addressing my comments in detail. I am fully satisfied with their responses to my and Reviewer 2's points, and have no further comment. Congratulations on an excellent study.

Appendix A

The manuscript is a welcomed addition of results to a highly relevant area for management and conservation. Much faith and hope is currently being put into agent based modelling to supply answers needed for tough decisions, but as always with models the quality of these answers depends critically on the quality of the input data.

Overall the study confirms results from previous studies with similar setups, showing reaction distances to pile driving in the range of tens of kms, but adds to our understanding of the temporal development of the response (possible habituation) and relative importance of the different noise sources (pile driving and ADDs).

The study is well designed. While there is no reason to expect issues with analysis of sound recordings and modelling of sound exposures, not enough details are provided to judge this. Same could be said about the C-POD data, but here analysis seems to follow de facto standards for this type of studies. There are some issues with interpretation if the results, as outlined in the detailed comments below.

line 13. The word dose-response curve should be reserved to responses in individual animals. As the authors state themselves in the main text, the reduced reaction distance with time could be due to a displacement of more sensitive individuals (a form of survivor bias), which means that the dose-response curves of the individuals could remain constant, but the composition of animals change.

line 75. It is clear that ADDs can cause reactions in porpoises at tens of km distance, but no-one has shown or even suggested that vessel noise could have such far-reaching effects.

Line 102-108. Details about analysis of the acoustic measurements are missing. What acoustic metrics were quantified and with what temporal resolution? How much of the data was analysed? All of it, or smaller snippets considered statistically representative? If the latter, how was that justified? information about the modelling of sound propagation is lacking completely. Which type of model was used and what software implementation? What was the frequency resolution of the modelling and what environmental layers were used? All of this must be described in some detail (perhaps in supplementary material), if the reader is going to have any faith in the modelled map on figure 2. How was the measured data used in the modelling and how well was the model able to predict measured levels?

Line 113-116. How was the audiogram-weighting function derived from Kastelein et al 2002 different from the generalised NOAA-curve (derived from the same data)? Was the uncorrected data from Kastelein 2002 used, or the corrected data from Kastelein et al (2010)? The uncorrected 2002 data are wrong and should not be used. A curve or an equation for the audiogram-weighting should be given, in order for others to replicate the analysis.

Line 141-146 (and later). It is an inherent limitation of studies with this design that distance to source is confounded with received source level. This is mentioned by the authors, but perhaps should be made even more clear to the reader.

Line 186-189. Although mentioned later in the discussion, it should be clear already here that a decrease in porpoise detections in the presence of vessels could indicate either a response of porpoises, a masking of detections of the C-POD by the vessel noise, or both. At the moment, it is not possible to tell the two effects apart and great care should be taken in interpretation of these results.

Line 194. As stated above, there are two issues with using dose-response curve in this context: the distance-dose ambiguity and the fact that you are not evaluating responses of individuals, but evaluating the proportion of the local population that reacts.

Line 198-200. There are at least two problematic issues with the interpretation of the ADD result:

1. There is only one exposure of each type, which means that from a purist point of view the conclusion is not valid. Strictly, it can only be said that the animals reacted less to piling #62 than #61, but without replicates it could be a coincidence. I don't think so, but you have not demonstrated it convincingly. See Dähne et al (2017) for support for the conclusion, but in any case you must argue more convincingly. Even though there are no replicate exposures, there are replicate measurements. They are pseudoreplications in relation to the stimulus (you don't know whether it really is the ADD or something else), but true replicates with respect to the response itself.
2. When comparing the noise level from the two exposures are you then evaluating the weighted level of the ADD signal or the pile driving noise? If the assumption is that they respond to the ADD, it should be the level of that sound which is used, not the pile driving noise. In any case, it would be useful to have both measures of ADD and pile driving noise.

Line 207-208. This sentence does not really make sense, for several reasons. First, dose-response (in my opinion) relates to individuals, not populations. Such dose-response relationships can only be determined by observation, and serve as input to the models, not output. Second, there can be legitimate reasons for managing based on habitat loss (even if temporary), when it comes to protected or otherwise sensitive areas. For those areas minimizing habitat loss may be a goal in itself, even if the long-term survival of the population is not likely to be compromised directly.

Line 214. Why potential disturbance? You (as many others before you) are measuring actual disturbance, not potential.

Line 223-227. You should also consult the studies on bottlenose dolphins in Shark Bay, e.g. Bejder, L., A. Samuels, H. Whitehead, and N. Gales. 2006. Interpreting short-term behavioural responses to disturbance within a longitudinal perspective. *Animal Behaviour* 72:1149-1158. Responses in naive porpoises from Moray Firth may or may not be transferable to porpoises in the southern North Sea. Little is known about both populations, but differences in prior exposure (as mentioned), but also how local or wide roaming the populations are, are important to keep in mind when transferring conclusions from one area to the other.

Line 252-258. Why is it less likely that they change vocal behaviour?. There are results from German wind farms that show that porpoise detections reappear close to the piling size so fast after end of the pile driving that it is impossible that they could have swam back. They must have been there all the time, just less detectable acoustically (and from aerial surveys). Anyways, cessation of vocalisations is as much a response as fleeing, so the distinction does not invalidate the results. One can in fact argue that the impact on the animals is larger if they remain more or less motionless in the area, compared to fleeing to somewhere else, where they can resume feeding.

Line 262. How can you claim that $5E-4$ animals/km² is a relatively high density? This is a density corresponding to less than 5 porpoises in the entire Moray Firth. Detection conditions in the Baltic are very different from the North Sea, not just because of fewer animals, but also because of the low salinity. The logic simply isn't there

and there is no experimental evidence to support the claim that environmental conditions did not affect detection probabilities. The key question is whether any effects could be large enough to affect your conclusions.

Line 311-312. Do not add new results in the discussion. Please describe these results in the results section together with the other ADD results.

Line 318-322. It is important that you qualify this statement about vessel noise further. Are you implying that porpoises responded to the vessels around the piling site at distances comparable to the pile driving noise and the ADD? Or is this a local response to service boats to and from the wind farm, passing by the C-PODs far away from the wind farm? The first claim requires substantially better support from data than what you have presented. The second claim may be valid, but less relevant to the discussion of effects of the pile driving per se, and more relevant for a discussion of the construction phase as a whole. And there is the issue of masking of C-PODs by ship noise.

Line 326-332. Because distance and received level are confounding factors, it is difficult to conclude on the frequency weighting. All weighted levels are likely to be highly correlated and the models really does not allow for separation between models using distance (AIC 619), Audiogram (AIC 620), NOAA (AIC 623), and M-weighting (AIC 625). These differences must be said to be marginal and inconclusive. I'm not arguing against audiogram weighting, but this dataset is not well suited to test this question.

Appendix B

We thank both Reviewers for all their time and effort in reviewing our paper. Although the comments were extensive, they were supportive and entirely constructive. We therefore found ourselves in an unusual position where we were genuinely happy to take action on all the comments, and we appreciate the improvement the comments should have made to the manuscript.

Underlined text (in this Response) indicates the action we have taken in response to the Reviewers' comments. We have uploaded a version of the manuscript with the changes tracked (Graham_BOWL_cMMMP_Porpoise_piling_responses_MS_revised_changes tracked_300419.docx): where text has been added, altered or deleted this is indicated by a comment in the track changes manuscript with the reviewer comment number e.g. R1.1.

Reviewer 1	Response
(R1.1) Throughout the manuscript, I found that indications of the uncertainty in model results and predictions were lacking. For example, the distances at which the expected response is 50% or the percentages of individuals displaced within a 26-km radius are presented as single numbers. This carries the danger of these numbers being used as fixed truths in future policies (just as the 26 km radius is used now). Rather than reporting single expected values, I would refer to the confidence intervals.	We agree with this comment and have now reported 95% confidence intervals for all distances and noise levels at which the expected response is 50% both in the Abstract and the Results section, as well as for the number and percentage of individuals displaced within a 26-km radius in the Abstract and the Discussion.
(R1.2) I found the description of the approach used to define a response slightly confusing (as I detail below). Overall, I like the solution the authors have devised, but it took a while to wrap my head around it, so I suggest rethinking the presentation of the methods to improve clarity (below I suggest using a sample timeline with the indication of the various periods).	Figure 3 with a sample timeline has been added and text inserted to improve clarity (see also response to comments below).
(R1.3) Finally, I have some concerns on the use of proportional changes in occurrence as a measure of response. This should in theory account for any seasonal variability in occurrence, but, if occurrence is very low in some periods of the year, proportions likely become noisier, and a temporal trend in the response becomes harder to tease apart from the temporal trend in occurrence (see also my comment below on the short reference window used). Another complication of using proportional change is that it cannot disentangle permanent changes in occurrence; in other words, it is currently assumed that a 96-h interval is sufficient for the occurrence to return to baseline, but porpoises could be leaving and not returning. Remaining individuals may be less responsive (as the authors briefly discuss) and more responsive individuals could be simply kept away from these locations because the interval between events is not sufficient for them to return. This would therefore not be an indication of diminishing response or tolerance/habituation (as implied in most of the manuscript, including the title), but rather of an intense, prolonged effect. I think it is worth checking whether there is any indication of these processes occurring, for example by repeating the analysis only using events separated by longer gaps (192 h?), or investigating any	In response to the Reviewer's concerns we have added a figure (Figure 5) showing the temporal trend in porpoise detections through 2017 within the construction site. This demonstrates that there was no evidence of a long-term decline in occurrence (including the baseline periods) during the construction period. We have added text at the start of the Results section to clarify this point. In addition, we repeated the analysis of the 24-h response with distance using only those events separated by longer gaps (192 h), which did not change the model. Parameter estimates were very similar to those for the model derived using the larger dataset for events preceded by a shorter break in piling. We have added text to the Results and Table S4 to the Supplementary material.

temporal trend in occurrence in baseline periods (and comparing this trend with the trend in responses). These checks could be included in the Supplementary material.	
Detailed comments	
(R1.4) Line 16: I would clarify that ‘location’ refers to the location of piles being driven; this only became clear after reading the Methods section.	We have inserted the word “piled” to make this clearer.
(R1.5) Line 37: Please cite the recent review of the population consequences of disturbance framework (Pirootta et al. 2018, Ecology and Evolution, https://doi.org/10.1002/ece3.4458).	Done.
(R1.6) Line 42: You could cite the work by Miller et al. 2014 (JASA, doi: 10.1121/1.4861346) on killer whale dose-response analysis. Also, it is worth mentioning the study by Moretti et al. 2014 (PLoS One; doi: 10.1371/journal.pone.0085064) in this section; they have also used passive acoustics to quantify group-level risk functions.	These 2 studies are now cited.
(R1.7) Lines 85-90: You currently do not list your assessment of the effects of shipping traffic among the aims, which I find as important as the ADD analysis.	We have inserted text to these lines to refer to our assessment of the effects of shipping traffic.
(R1.8) Lines 113-116: You may be aware that updated weighting functions have recently been published by Southall et al. (2019; Aquatic Mammals, doi: 10.1578/AM.45.2.2019.125). I don’t think it’s necessary to rerun the entire analysis, but it would be interesting to compare how the weighting differs in the updated version.	We are aware of these weighting functions, and that they are identical to the previously published NMFS (2018) weighting functions used in the analysis. We have added text to explicitly refer to this paper in the methods.
(R1.9) Lines 126-129: A few aspects are unclear in this section: 1) How did you compute DPH (which you define as detection positive hours per day) over a 12-h period? 2) Why is the 48-h baseline period calculated from the start of piling for 24 h analysis, and from the end of piling for the 12 h analysis? This seems like an important detail but I struggle to see a justification. 3) Do I understand correctly that a 12-h period is used as a baseline for the 12-h analysis, and a 24-h period for the 24-h analysis? In general, it might help to have a schematic representation of the timeline for one piling event, identifying the two response and the two baseline periods.	1. The definition of DPH should be “detection positive hours”, the inclusion of the words “per day” was an error and “per day” has now been deleted from this line. DPH was therefore computed either for a 12-h or 24-h period, which should now be clear. 2. The reason for choosing different start times for the baseline periods relates to balancing the need to match the baseline and response periods (for the 12-h response) with respect to the diel cycle while maximizing the time between the end of the baseline period and the start of piling to avoid overlap of the baseline with potentially disturbing activities such as vessel anchoring. This has now been explained by adding text to this section of the Methods. 3. Yes this is correct, and has been clarified by including Figure 3 showing a schematic representation of the timeline for one piling event.
(R1.10) Line 135-136: The description of how this baseline variation was computed and used only became clear after reading the Supplementary material. I would consider bringing that section (Figure S1) into the main text, for clarity. Also, I understand your rationale, but this approach implicitly assumes that the proportional change	Whether the proportional change in occurrence varies seasonally or in relation to the response/baseline period (24-h vs 12-h) is a valid concern. There were few breaks in piling of sufficient length to examine variation in the proportional change in occurrence from April to the end of

in occurrence in those two weeks in March is representative of proportional changes over an entire year. This could be true, but I could also imagine a scenario where the proportional change in occurrence varies seasonally (if, for instance, porpoises use the area more transiently in one season and more consistently in another). I think this assumption is quite important, and should be mentioned in the Discussion. Also, the baseline proportional change is only calculated for a 24-h period. I'd be curious to know if the 99th percentile change remains the same when considering 12-h windows.	November (through the period of active pile driving) and such breaks as there were, were not distributed evenly in time through this period. Therefore in response to this comment we randomly sampled a similar two week period in December after all pile driving had ceased, and calculated the 1st percentile for the March and December data combined. The value of the 1st percentile (of the distributions for the 24-h and 12-h periods) was then calculated. This did not change the threshold used for the 24-h response but did give a slightly different threshold for the 12-h response (-0.667 rather than -0.5). This 1st percentile for the 12-h response/baseline is lower than the original threshold used to define a response (-0.5), meaning that model predictions would be less conservative if we used this lower threshold. To be both consistent and precautionary, we therefore argue that it is most appropriate to use the same -0.5 threshold for both the 24-h and the 12-h response. Figure S1 has been updated with the new data, and both the figure and the methods description have been moved into the main text. In addition, we have added text to the discussion to mention the potential implications of threshold choice.
(R1.11) Line 146: From Table S1 and the Discussion, I gather there was only one event where no ADD was used. I agree this is still an important piece of information, and it should be reported. However, with one sample, you cannot rule out the option that some other confounding factor has contributed to porpoises reacting less on that day. I think you should discuss this carefully and tone down your statements regarding the use of ADDs throughout the manuscript. This is an important preliminary observation, but at the moment it is highly correlative.	In response to this comment, and Reviewer 2's comments below (R2.8), in the Results section we have now referred to the additional analysis that we carried out including a second event where no ADD was used (originally only detailed in the ESM, table S3). We agree that our results are constrained by limited sample sizes, which we have stated at the beginning of the Discussion "Despite smaller sample sizes, there was preliminary evidence that shorter-term responses to the cumulative impact of ADD and impact piling were greater than responses to pile driving alone" but we have inserted the word "preliminary" to highlight the need for further work and the limitations of our study in this respect. We have also expanded the discussion to clarify the limitations of this result.
(R1.12) Line 149: I assume that by 'response period', you mean the 12-h and 24-h windows, and not the period of piling, correct?	Yes, this has been clarified by adding the text, 12-h or 24-h, to this line.
(R1.13) Line 152: You should probably test for any remaining autocorrelation in model residuals.	We did plot the model residuals to check for any remaining autocorrelation – there was none. We have inserted text to the Methods to outline the checks we performed on model residuals.
(R1.14) Line 154: Did you use REML to fit your mixed-effects models in lme4? If so, you cannot use AIC to compare models with different fixed-effects components (see here, for example:	Models were fitted using maximum likelihood not REML, so this shouldn't be an issue.

https://stats.stackexchange.com/questions/131272/lme4-why-is-aic-no-longer-displayed-when-using-rem1). The models must be refitted using ML for comparison between different fixed-effect structures (the function extractAIC should do that automatically). In my experience, results often do not change much.	
(R1.15) Line 175: ‘Highly significant’ implies the use of some hypothesis test, of which I could not find any mention in the text. Also, if this is indeed statistical significance, I would remove the word ‘highly’: after choosing a significance level, a result is either significant or not.	The word ‘highly’ has been deleted from this line. In addition, text has been added to the Methods section and to the Results to document the test and test results used to determine significance of this covariate.
(R1.16) Line 216: Again, I suggest citing Moretti et al. 2014 here.	Done.
(R1.17) Line 269: And I would cite Pirotta et al. 2018 here.	Done.

Reviewer 2	Response
(R2.1) Line 13. The word dose-response curve should be reserved to responses in individual animals. As the authors state themselves in the main text, the reduced reaction distance with time could be due to a displacement of more sensitive individuals (a form of survivor bias), which means that the dose-response curves of the individuals could remain constant, but the composition of animals change.	We recognize this limitation in our study and in response to the Reviewer’s comment we have altered the wording in line 13 to remove the reference to dose-response curve that implies responses in individual animals. Elsewhere in text, we have either inserted the word proxy before dose-response or changed the wording to “behavioural response function”.
(R2.2) Line 75. It is clear that ADDs can cause reactions in porpoises at tens of km distance, but no-one has shown or even suggested that vessel noise could have such far-reaching effects.	We are not suggesting this either, so we have deleted the text “and vessels” from this line to avoid this confusion.
(R2.3) Line 102-108. Details about analysis of the acoustic measurements are missing. What acoustic metrics were quantified and with what temporal resolution? How much of the data was analysed? All of it, or smaller snippets considered statistically representative? If the latter, how was that justified? Information about the modelling of sound propagation is lacking completely. Which type of model was used and what software implementation? What was the frequency resolution of the modelling and what environmental layers were used? All of this must be described in some detail (perhaps in supplementary material), if the reader is going to have any faith in the modelled map on figure 2. How was the measured data used in the modelling and how well was the model able to predict measured levels?	We have now included a detailed description of the model and the corroboration with measured data in the Supplementary Material, which addresses each of the Reviewer’s queries. This includes tabulated data (Table S1) on the agreement between the model and the measurements.
(R2.4) Line 113-116. How was the audiogram-weighting function derived from Kastelein et al 2002 different from the generalised NOAA-curve (derived from the same data)? Was the uncorrected data from Kastelein 2002 used, or the corrected data from Kastelein et al (2010)? The uncorrected 2002 data are wrong and should not be used.	We thank the Reviewer for this correction, and have rerun the analysis with the updated audiogram (Kastelein et al 2010) accordingly. This did not significantly alter the results or conclusions from the analysis. Figure 6b, the parameter estimates in Table 1 and the predictions from the model in the Results have been updated with the revised figures. A figure showing

A curve or an equation for the audiogram-weighting should be given, in order for others to replicate the analysis.	the audiogram weighting has also been included in the Supplementary Material (Figure S1).
(R2.5) Line 141-146 (and later). It is an inherent limitation of studies with this design that distance to source is confounded with received source level. This is mentioned by the authors, but perhaps should be made even more clear to the reader.	This limitation has been clarified now at the end of the discussion (also in response to R2.17).
(R2.6) Line 186-189. Although mentioned later in the discussion, it should be clear already here that a decrease in porpoise detections in the presence of vessels could indicate either a response of porpoises, a masking of detections of the C-POD by the vessel noise, or both. At the moment, it is not possible to tell the two effects apart and great care should be taken in interpretation of these results.	We have inserted text following on from these lines in the last paragraph of the results to make this clear here.
(R2.7) Line 194. As stated above, there are two issues with using dose-response curve in this context: the distance-dose ambiguity and the fact that you are not evaluating responses of individuals, but evaluating the proportion of the local population that reacts.	We have re-worded this line to take account of the Reviewer’s concerns.
Line 198-200. There are at least two problematic issues with the interpretation of the ADD result:	
(R2.8) 1. There is only one exposure of each type, which means that from a purist point of view the conclusion is not valid. Strictly, it can only be said that the animals reacted less to piling #62 than #61, but without replicates it could be a coincidence. I don’t think so, but you have not demonstrated it convincingly. See Dähne et al (2017) for support for the conclusion, but in any case you must argue more convincingly. Even though there are no replicate exposures, there are replicate measurements. They are pseudoreplications in relation to the stimulus (you don’t know whether it really is the ADD or something else), but true replicates with respect to the response itself.	See also response to Reviewer 1’s comment above on Line 146 (R1.11) that raises similar concerns about the ADD result. There was (in the main analysis where a 4-day gap in piling was required) only 1 exposure to piling without ADD mitigation. However, there were 16 exposures to piling with ADD mitigation, and where we allowed a slight relaxation of the length of break between piling events, we were able to include a second exposure to piling without ADD mitigation. Clearly our sample size still constrains interpretation of the result but we have expanded discussion of the limitations of our sample size (see also response to earlier comment – R1.11) and have concluded the paragraph in the discussion, where we discuss the limitations of the result with the statement “but further work is required to better understand the relative contribution of pile driving, ADD noise and vessel activity to observed responses of cetaceans to offshore construction”. We do also cite Dähne et al (2017) in our discussion of the contribution of ADDs.
(R2.9) 2. When comparing the noise level from the two exposures are you then evaluating the weighted level of the ADD signal or the pile driving noise? If the assumption is that they respond to the ADD, it should be the level of that sound which is used, not the pile driving noise. In any case, it would be useful to have both measures of ADD and pile driving noise.	We have not compared noise levels from exposures with and without ADD due to the difficulties of integrating noise levels from both impulsive and continuous sources. Instead, we have considered responses to pile driving noise alone, as this is what is currently considered the major source of disturbance in impact assessments. However, by including ADD activity as a factor in these models, we highlight that other noise sources should be investigated in more detail. We have now removed the second panel from

	Figure 7 and the model with audiogram-weighted noise levels from Table 1 to avoid any potential confusion over this point. We aim to explore this further in a second publication that will present measurements of ADD noise from, and porpoise responses to, experimental ADD exposure in the absence of piling.
(R2.10) Line 207-208. This sentence does not really make sense, for several reasons. First, dose-response (in my opinion) relates to individuals, not populations. Such dose-response relationships can only be determined by observation, and serve as input to the models, not output. Second, there can be legitimate reasons for managing based on habitat loss (even if temporary), when it comes to protected or otherwise sensitive areas. For those areas minimizing habitat loss may be a goal in itself, even if the long-term survival of the population is not likely to be compromised directly.	We have altered the wording of this line to take account of the Reviewer’s comment. In addition, we have inserted text later in the paragraph to take into consideration the Reviewer’s second point.
(R2.11) Line 214. Why potential disturbance? You (as many others before you) are measuring actual disturbance, not potential.	Agreed – we have deleted the word “potential” from this line.
(R2.12) Line 223-227. You should also consult the studies on bottlenose dolphins in Shark Bay, e.g. Bejder, L., A. Samuels, H. Whitehead, and N. Gales. 2006. Interpreting short-term behavioural responses to disturbance within a longitudinal perspective. Animal Behaviour 72:1149-1158. Responses in naive porpoises from Moray Firth may or may not be transferable to porpoises in the southern North Sea. Little is known about both populations, but differences in prior exposure (as mentioned), but also how local or wide roaming the populations are, are important to keep in mind when transferring conclusions from one area to the other.	We have added in this citation and text to these lines and added a point to the discussion about the need to consider residence patterns.
(R2.13) Line 252-258. Why is it less likely that they change vocal behaviour? There are results from German wind farms that show that porpoise detections reappear close to the piling size so fast after end of the pile driving that it is impossible that they could have swam back. They must have been there all the time, just less detectable acoustically (and from aerial surveys). Anyways, cessation of vocalisations is as much a response as fleeing, so the distinction does not invalidate the results. One can in fact argue that the impact on the animals is larger if they remain more or less motionless in the area, compared to fleeing to somewhere else, where they can resume feeding.	We agree that cessation of vocalisation is as much a response as fleeing and have added text to make this point and also the Reviewer’s final point that potentially the impact of staying (and remaining silent) could be more severe than fleeing. It is simply the length of time, hours rather than minutes, which lead us to suggest that it seems less likely that the decrease in detections is due to a change in vocal behaviour, however clearly this is a possibility that we cannot discount – we have attempted to clarify this in the text.
(R2.14) Line 262. How can you claim that 5E-4 animals/km² is a relatively high density? This is a density corresponding to less than 5 porpoises in the entire Moray Firth. Detection conditions in the Baltic are very different from the North Sea, not just because of fewer animals, but also because of the low salinity. The logic simply isn’t there and there is no	We were not claiming that 5E-4 animals/km² was a high density, however we concede the Reviewer’s point here. We have amended this paragraph accordingly, and additionally made the point that we originally intended to make that the use of our response variable should have accounted for any spatial or temporal variation in detection probability.

experimental evidence to support the claim that environmental conditions did not affect detection probabilities. The key question is whether any effects could be large enough to affect your conclusions.	
(R2.15) Line 311-312. Do not add new results in the discussion. Please describe these results in the results section together with the other ADD results.	We have now described the results of the additional analysis that we carried out including a second event where no ADD was used in the results section (details are included in the ESM, table S3).
(R2.16) Line 318-322. It is important that you qualify this statement about vessel noise further. Are you implying that porpoises responded to the vessels around the piling site at distances comparable to the pile driving noise and the ADD? Or is this a local response to service boats to and from the wind farm, passing by the C-PODs far away from the wind farm? The first claim requires substantially better support from data than what you have presented. The second claim may be valid, but less relevant to the discussion of effects of the pile driving per se, and more relevant for a discussion of the construction phase as a whole. And there is the issue of masking of C-PODs by ship noise.	We have further qualified this statement to take account of the Reviewer’s concerns by adding text. This should clarify that we view this as a local response to service boats to and from the wind farm that could represent an important contribution to the cumulative impact of the construction phase as a whole, but not on the same spatial scale as pile driving or ADDs. We have also re-iterated the possibility that vessel noise could have masked porpoise detections.
(R2.17) Line 326-332. Because distance and received level are confounding factors, it is difficult to conclude on the frequency weighting. All weighted levels are likely to be highly correlated and the models really does not allow for separation between models using distance (AIC 619), Audiogram (AIC 620), NOAA (AIC 623), and Mweighting (AIC 625). These differences must be said to be marginal and inconclusive. I’m not arguing against audiogram weighting, but this dataset is not well suited to test this question.	We agree our study was limited in its ability to test this question and in order to emphasize this point, we have added text after these lines to highlight this limitation. We do then go on to say a couple of lines later “In future, integration of data from multiple windfarm sites could provide a wider range of received noise levels at given distances from piling, helping disentangle the relative importance of distance to source and loudness in shaping behavioural responses”.